# TacticZero: Learning to Prove Theorems from Scratch with Deep Reinforcement Learning

**Minchao Wu**
Australian National University
Data61, CSIRO
`Minchao.Wu@anu.edu.au`

**Michael Norrish**
Australian National University
`michael.norrish@anu.edu.au`

**Christian Walder**
Australian National University
Data61, CSIRO
`christian.walder@data61.csiro.au`

**Amir Dezfouli**
Data61, CSIRO
`amir.dezfouli@data61.csiro.au`

## Abstract

We propose a novel approach to interactive theorem proving (ITP) using deep reinforcement learning. The proposed framework is able to learn proof search strategies as well as tactic and arguments prediction in an end-to-end manner. We formulate the process of ITP as a Markov decision process (MDP) in which each state represents a set of potential derivation paths. This structure allows us to introduce a search mechanism which enables the agent to efficiently discard (predicted) dead-end derivations and restart from promising alternatives. We implement the framework in the HOL4 theorem prover. Experimental results show that the framework using learned search strategies outperforms existing automated theorem provers (*i.e. hammers*) available in HOL4 when evaluated on unseen problems. We further elaborate the role of key components of the framework using ablation studies.

## 1 Introduction

Interactive theorem proving (ITP) is the process of humans interacting with a computer system to develop formal proofs of mathematical theorems. In ITP, a human user can define mathematical objects, and then guide the computer to prove theorems about them using commands called *tactics*. Tactics are programs that embody high-level proof strategies such as simplification and induction. Successful tactic application converts the current target *goal* (a theorem to be proved) into zero or more subgoals that remain to be proved.

Proofs written in an ITP system are ultimately checked by a machine, and therefore, they are much more trustworthy than pencil-and-paper proofs. For this reason, ITP has gained success in both formalizing mathematics, and in verifying computer programs [Leroy, 2009, Hales et al., 2017]. Nevertheless, because a great amount of formal detail needs to be spelled out when using an ITP system, substantial human inputs are required to fill the gaps between steps in a proof. In particular, in terms of tactic-based theorem proving, human guidance in the selection of both tactics, *and* the arguments to tactics is crucial to a successful proof. This requires expert knowledge both of the relevant mathematics, and the particular ITP system being used. This requirement for expertise has in turn limited the application of ITP systems.

One promising line of work to address this limitation has been to use machine learning methods to replace the role of human experts in ITP. Existing learning-based approaches have two noticeable

characteristics: learning from human examples for tactic prediction and using a fixed proof search strategy (*e.g.* breadth/best-first search (BFS)) that is separate from the learning process to find proofs. The dependency on human examples limits the application of this kind of techniques depending on the availability and quality of the human proofs in the domain that the proof is required. There are potentially infinitely many proofs of a theorem, but there are only one or two of them exist in the library without any gurantee of being optimal. For this reason, it is essential for an agent to be able to explore different proofs by itself in order to generalize well and capture the underlying mathematical knowledge. On the other hand, fixed proof search strategies tend to be artificial and suboptimal. Search strategies such as BFS are usually computationally expensive in time and space (due to needless proof states resulting from redundant expansion of previous proof states), and are thus not suitable for learning algorithms that require fast iteration.

We propose a reinforcement-learning (RL) framework to learning ITP that addresses the above limitations of the existing approaches. The RL agent interacts (Figure 1a) with the HOL4 interactive theorem prover [Slind and Norrish, 2008] and mimics the behavior of human ITP experts. Given a proof state, a human expert often decides not only what tactic and arguments to apply, but also whether or not to go back to a previous proof state and restart the proof from there. This kind of situation occurs frequently in ITP as careless tactic applications can quickly result in unmanageable proof states. An experienced ITP expert would have the intuition on when and where to backtrack. Our agent learns proof search strategies in a similar manner by learning to choose promising derivations and subgoals to attack, as well as the tactics and their arguments to apply to the goals, without using the limited examples in the library.

**Our Contributions** are as follows.

- We provide a formulation of ITP as a Markov decision process (MDP) [Bellman, 1957] in which flexible state representations enable tracking multiple subproofs during the search process. The structure of the action space allows for tactics to have both theorem names and terms as input arguments.

- We propose an RL architecture using multiple recurrent and feed-forward neural network modules to solve the above MDP. We use policy gradients [Williams, 1992] to jointly learn to apply actions at the backtracking, goal, tactic and argument levels.

- We implement the framework in the HOL4 theorem prover and show that it outperforms state-of-the-art automated theorem provers (*i.e. hammers* [Gauthier and Kaliszyk, 2015]) available in HOL4 including E [Schulz et al., 2019], Z3 [de Moura and Bjørner, 2008] and Vampire [Kovács and Voronkov, 2013]. We further use ablation studies to establish the contribution of key components of our architecture in achieving the overall performance.

## 2 Interactive Theorem Proving with Reinforcement Learning

We begin with an overview of interactive proving in HOL4 by a simple example (corresponding to the script in Figure 2). Suppose we want to prove the theorem $p \land q \Rightarrow p \land q$ where $p$ and $q$ are propositional variables (*i.e.*, boolean). This expression is called our initial *goal*. Since it is an implication, we will assume the antecedent, and then use that to prove the conclusion. This can be done by applying a *tactic*, rpt strip_tac to the goal. This tactic does not take any arguments.

```
Theorem example:
  p ∧ q ⇒ p ∧ q
Proof
  rpt strip_tac
  >- (* Induct_on 'p' *) simp[]
  >- simp[]
QED
```

Figure 2: A Simple HOL4 Proof.

HOL4 tells us that the tactic application results in two new *subgoals*. In this case, they are $p \Rightarrow q \Rightarrow p$ and $p \Rightarrow q \Rightarrow q$.[1] At this stage, both subgoals need to be proved in order for the proof to succeed. Suppose we choose to first prove $p \Rightarrow q \Rightarrow p$. We notice that because $p$ and $q$ are boolean variables, it should be possible to prove the goal by cases. This can be done by applying the Induct_on tactic to the goal. The Induct_on tactic takes a single *term* argument, that is, an arbitrary HOL4 expression. In this case, it will be the variable $p$.

---

[1]HOL4 actually moves $p$ and $q$ to what it calls the goal's *assumption-list*; for simplicity here, we represent that list with chained implications in one formula.

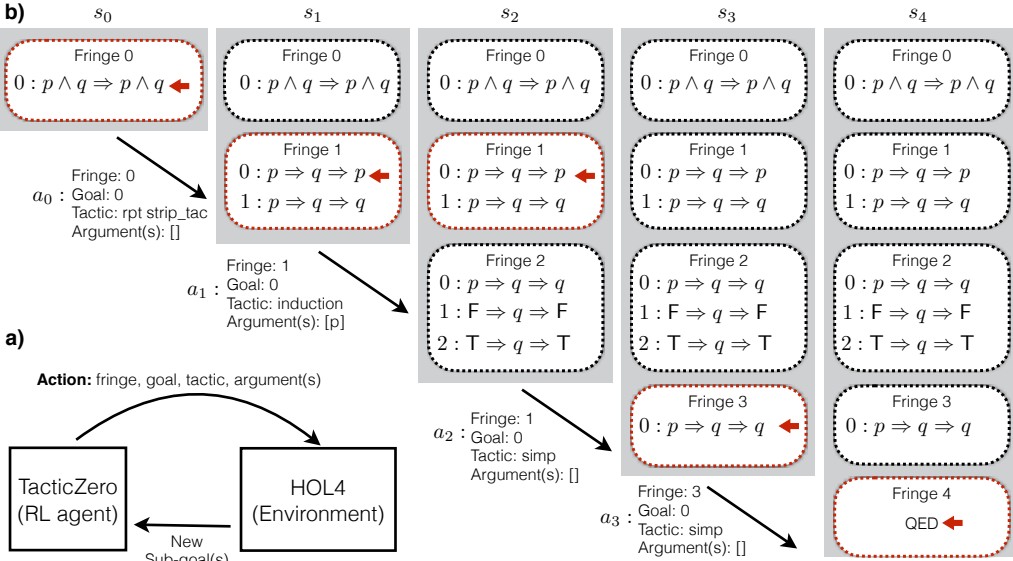

Figure 1: *a)* Interaction between the RL agent and the environment. *b)* An example scenario. An action ($a_i$) consists of a fringe, a goal, a tactic and its arguments. Each state consists of multiple fringes and each fringe includes a set of goals that are sufficient to be proved to derive the original target theorem. At each state, the agent evaluates goals within each fringe and decides which fringe is more promising, and within each fringe which goal to work on first (shown by red arrows). Given this choice, the agent then selects a tactic and appropriate arguments. At each state the agent can cease working on the newly added fringe, and backtracks to an older fringe if the new one is predicted to be hard. This process continues until an empty fringe is derived ($s_4$).

After applying `Induct_on` 'p' to $p \Rightarrow q \Rightarrow p$, two more subgoals are generated, with the variable $p$ replaced by truth values $\mathsf{T}$ and $\mathsf{F}$ respectively. At this stage, we have three subgoals to deal with. One is the earlier goal that we chose to defer: $p \Rightarrow q \Rightarrow q$. The other two are the newly generated $\mathsf{T} \Rightarrow q \Rightarrow \mathsf{T}$ and $\mathsf{F} \Rightarrow q \Rightarrow \mathsf{F}$.

One might decide at this stage that the previous tactic application of `Induct_on` 'p' has made the situation worse because it has left us with more goals to prove. A human expert may now look back at the remaining subgoals before the application of `Induct_on` 'p' and try another tactic. Fortunately, there is a tactic called `simp` that can prove the goal $p \Rightarrow q \Rightarrow p$ directly. The `simp` tactic takes a list of theorem names as its argument. In this case, an empty argument list is sufficient to prove the goal. Similarly, `simp[]` proves the remaining $p \Rightarrow q \Rightarrow q$ goal as well.

## 2.1 Modeling ITP with MDP

Tactic-based theorem proving is naturally understood as a tree search problem where a tree is composed of sets of goals and can be updated by applying tactics to a node. Below we give a formulation that reflects such intuition — states are the search trees and actions are the tactic applications to nodes.

**States** A proof attempt starts with a main goal $g \in \mathbb{G}$. At any point in a proof attempt, there is a set of goals that all need to be proved. We refer to these finite sets of goals as *fringes*. In the context of our framing as an MDP, multiple fringes will be generated and maintained in such a way that the main goal will be proved if everything in any one fringe is proved. Thus a fringe represents a particular path along which we may continue in an attempt to complete a single valid HOL4 proof of the main goal. In contrast, always choosing the most recently generated fringe would be equivalent to never *backtracking* in HOL4. The MDP state $s$ is therefore a finite sequence of fringes and we denote the $i$-th fringe in state $s$ by $s(i)$. See Figure 1b for an example state sequence.

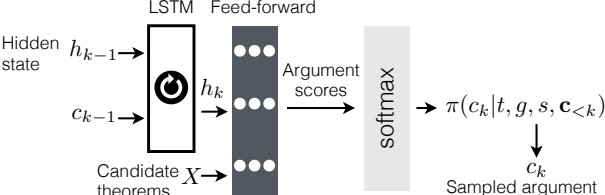

Figure 3: The argument network. The aim is to select arguments from candidates theorems $X$ for a tactic $t$ to be applied on goal $g$. To achieve this, the hidden state of an LSTM neural network are initialized using $g$. The LSTM layer takes the previously chosen argument as an input ($c_{k-1}$) and through a feed-forward and a softmax layer generates the probability of selecting each candidate theorem for the next argument ($c_k$). $c_0$ is initialized to $t$.

**Actions** An action is a 4-tuple $(i, j, t, \boldsymbol{c}) \in \mathbb{N} \times \mathbb{N} \times \mathcal{T} \times \mathcal{A}^*$. Intuitively, given a state $s$, $i$ selects the $i$-th fringe in $s$ and $j$ selects the $j$-th goal within fringe $s(i)$. Then, $t$ is the HOL4 tactic that we select from the set $\mathcal{T}$ of possible HOL4 tactics, and $\boldsymbol{c}$ is the possibly empty list of arguments that accompany $t$ (see Figure 1b). An argument in $\mathcal{A}$ is either a theorem name, or a HOL4 term. The existence of arguments makes the action space arbitrarily large—terms may be arbitrarily defined, and there are thousands of theorems that can be given as arguments to the tactic. This necessitates an RL algorithm that can handle large action and state spaces, for which we select policy gradients [Williams, 1992].

**Rewards** Feedback is received from HOL4 after each tactic application. If successful, it generates a set of new (sub-)goals such that proving all of them proves $g$; if this set is empty, then $g$ is itself proven. A tactic application may fail, either with an error indicating that it is not applicable, or with a timeout caused by HOL4 exceeding a maximum computation time which we set. In terms of our framing as an MDP, different numeric rewards are associated with the different cases described above.

**MDP State Transition** A proof attempt always starts with a single main goal $g$, and so the initial state is a single fringe containing one element, $g$. Given a state $s$, performing an action $(i, j, t, \boldsymbol{c})$ may not change the state. This happens when the tactic times out, has no effect, or is not applicable. Otherwise, the application of the tactic generates a set of new subgoals. This set $G$ of new subgoals may be empty, indicating that goal $s(i)(j)$ is immediately proved by tactic $t$. In any case, a new fringe is then constructed by first copying fringe $s(i)$, and then replacing the goal $s(i)(j)$ with the new subgoals $G$. Then we construct a new state $s'$ by adding the new fringe to state $s$. Each state constructed in this way maintains all possible partial proof attempts. The size of each state is *linear* in the number of timesteps so that the manipulation and storage of states are efficient.

**Termination** The MDP terminates when an empty fringe is constructed, which implies that we can recover a proof of the main goal. In this case, the agent receives a positive reward. The process is also terminated if the timestep budget is exceeded, in which case the proof attempt is unsuccessful and the agent receives a negative reward. The state $s_4$ in Figure 1b is an example of a terminal state.

## 2.2 A solution to the MDP formulation

**Encoding HOL4 expressions** Each HOL4 expression is represented by a 256-dimensional vector, which is learned by a seq2seq autoencoder [Kingma and Welling, 2014] for sequences pre-trained on 4360 definitions and theorems (tokenized and represented in Polish notation) in the HOL4 library. For implementation, we use a customized version of the pytorch-seq2seq[2].

**Selecting fringes** Let $\mathbb{G} \subseteq \mathbb{R}^{256}$ be the space of goals; we learn a function

$$V_{\text{goal}} : \mathbb{G} \to \mathbb{R}, \tag{1}$$

which, intuitively, represents how easy a goal is to prove. Given a state $s$ we define the score $v_{s(i)}$ of its $i$-th fringe as

$$v_{s(i)} = \Sigma_{g \in s(i)} V_{\text{goal}}(g). \tag{2}$$

---

[2]`https://github.com/IBM/pytorch-seq2seq`

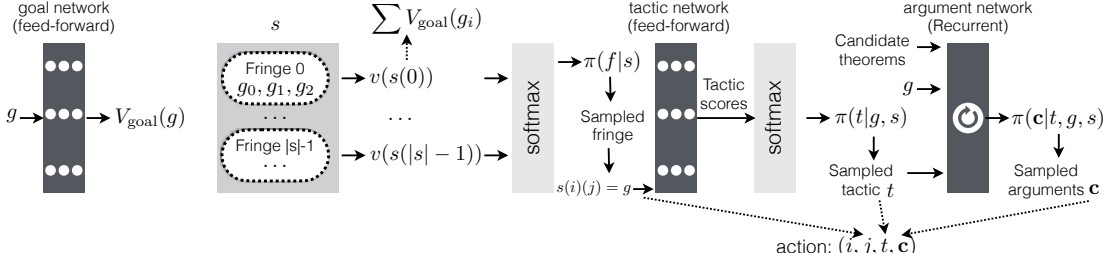

Figure 4: Structure of the model. The functionality of each network is described in subsection 2.2.

Summing in this way reflects a simplifying design choice; namely that *1)* to prove $s(i)$ we must prove all its consituent goals, *2)* that $V_{\text{goal}}$ outputs logits, and *3)* as we sum logits that the probability of solving each goal is independent given its numeric representation.

To choose a fringe to work on, the agent samples from the discrete distribution with probabilities

$$\pi_{\text{fringe}}(s) = \text{Softmax}(v_{s(0)}, ..., v_{s(|s|-1)}). \tag{3}$$

After selecting a fringe, by default we select the first goal in that fringe to work on, because all of the goals within a fringe have to be proved in order to prove the main goal, and the order in which they are proven is irrelevant.

**Selecting tactics** Suppose that we have selected a goal $g$ to work on. We learn a function

$$V_{\text{tactic}} : \mathbb{G} \to \mathbb{R}^D, \tag{4}$$

where $D$ is the total number of tactics allowed, which estimates the effectiveness of each tactic for proving $g$. To select the tactic, we again sample from the discrete distribution, here with probabilities

$$\pi_{\text{tactic}}(g) = \text{Softmax}(V_{\text{tactic}}(g)). \tag{5}$$

**Selecting a list of arguments** Suppose that we have selected a goal $g$ and a tactic $t$. If $t$ takes arguments, we need to select a list of them. This process is naturally modeled by a recurrent neural network architecture as shown in Figure 3.

The argument policy recurrence takes three pieces of data as its input. The first is a set of candidate theorems (or a set of candidate terms if $t$ takes a term) that the policy can choose from. The candidate theorems are usually the theorems proved earlier than $g$ in the library, and the candidate terms are the variables that occur in $g$. The second is the previously selected argument, which is initialized to the selected tactic. The third is a latent state variable which carries the sequential information of previously selected arguments over the argument generation process. This is because the arguments to tactics like `simp` generally depend on each other.

The first output of the recurrent module is a set of scores associated with the candidate arguments. We softmax these scores and sample to get one argument. An LSTM also produces a second output which is the latent state variable that carries the sequential information. The recurrent process is terminated when it reaches a pre-defined maximum length $L$ which we set for the argument list.

**Computing policy gradients** Given a state $s$, the selection process described above now allows us to define our policy distribution in terms of the following factors.

- $\pi(f|s)$, the probability of selecting fringe $f$ given $s$ induced by $\pi_{\text{fringe}}$ of Equation 3. Recall that, subsequently, the first goal $g$ in $f$ is always selected.

- $\pi(t|s, g)$, the probability of selecting tactic $t$ given $s$ and $g$, induced by $\pi_{\text{tactic}}$ of Equation 5.

- $\pi(c_l|s, g, t, \boldsymbol{c}_{<l}); l = 1, 2, \ldots, L$, the probability of selecting the $l$-th element of the argument list $\boldsymbol{c}$ given $s, g, t$ and the previously generated elements of the argument list. This is induced by the recurrent argument selection, where $L$ is the fixed length of arguments, and $\boldsymbol{c}_{<l} = (c_1, ..., c_{l-1})$.

Denoting the action by $a$, the policy therefore factorises as

$$\pi_\theta(a|s) = \pi(f|s)\pi(t|s,g)\prod_{l=1}^{L}\pi(c_l|s,g,t,\mathbf{c}_{<l}), \tag{6}$$

where $\theta$ represents the parameters of $\{V_{\text{goal}}, V_{\text{tactic}}, V_{\text{arg}}\}$. Denoting the $m$-th reward by $r_m$, and the MDP trajectory by $\tau = (s_0, a_0, r_0, s_1, a_1, r_1, \ldots, s_M, a_M, r_M)$, our objective function is the usual sum of discounted expected rewards with discount factor $\gamma \in (0, 1]$, *i.e.*,

$$J(\theta) = \mathbb{E}_{\tau \sim \pi_\theta}\Big[\sum_{m=0}^{M}\gamma^m r_m\Big]. \tag{7}$$

Despite the large action space at hand, optimisation in $\theta$ by gradient descent is made tractable by substituting the classic REINFORCE estimator [Williams, 1992] for $\nabla_\theta J$.

## 3 Experiments

### 3.1 Learning

**ITP settings** The learning task is to train an agent that is capable of proving theorems using a given set of tactics in HOL4's core library. The tactics allowed to be used by the agent are listed in Table 1. We collect[3] 1342 theorems that are known to be provable using the given set of tactics in HOL4's core library and randomly split them into training and testing sets in an 80:20 ratio. The training data merely provides the agent with the statement of theorems for it to learn to prove by itself. We set the timestep budget to be 50, and the execution time limit of each tactic application to be 0.1 seconds. Given a goal $g$, the candidate set of theorems that can be chosen as arguments to a tactic is all the theorems which come from the theories mentioned in $g$ and occur before $g$ in the library.

Table 1: Tactics that can be used by the agent.

| Tactics | Argument types |
|---|---|
| `drule`, `irule` | single theorem |
| `fs`, `metis_tac`, `rw`, `simp` | list of theorems |
| `Induct_on` | single term |
| `eq_tac`, `strip_tac` | none |

**Reward shaping** If a theorem is proved, a positive terminal reward depending on the difficulty of the target theorem is given. If the rate at which the theorem is being proved in earlier rollouts is above average, then the reward is 5. Otherwise, the reward is 15. If the proof attempt fails, the agent receives a terminal reward of -5. We also encourage the agent to "make progress" by giving it a 0.1 reward when new subgoals are generated and a 0.2 reward when a subgoal is solved. For all other outcomes after each timestep, a reward -0.1 is given. The negative terminal reward reduces the tendency of the agent to maximize the reward by making irrelevant progress that does not lead to a successful proof of the main goal. By nature of the tactics in Table 1, repeatedly applying the same tactic to the same goal will eventually result in unchanged fringes, in which case the -0.1 reward will be given.

**Replaying** One difficulty during training is that positive rewards are sparse. The agent may find a proof of a difficult theorem by accident, but then take many episodes to prove it again. To help the agent recall its successes, we maintain a replay buffer of earlier successful proofs of each theorem. During training, if the agent fails to prove a theorem it was previously able to prove, replaying will be triggered so that the agent is walked through one of the 5 most recent successful proof and parameters updated correspondingly. This represents a departure from pure policy gradients, but worked well in our experiments, presumably because the update remains in the high dimensional direction $\nabla_\theta J$ and therefore differs from a precise *on-policy* update (such as MAPO [Liang et al., 2018]) only by the ad-hoc choice of learning rate.

---

[3]See supplementary materials for a description of the method.

**Training** We jointly train the policies using RMSProp [Hinton et al., 2012] with a learning rate of $5 \times 10^{-5}$ for each policy. The discount factor $\gamma$ is set to be $0.99$. The structure of our model is illustrated in Figure 4. It takes two weeks with a single Tesla P100-SXM2 GPU and a single Intel(R) Xeon(R) CPU E5-2690 v4 @ 2.60GHz to complete 800 iterations.

## 3.2 Evaluation

In this section, we study the theorem proving capability of TacticZero by comparing it with three state-of-the-art automated theorem provers (Z3, E and Vampire) available as hammers in HOL4. When choosing arguments (premises), TacticZero and each hammer will use the same set of candidate theorems as described in subsection 3.1. For each hammer, we first call them with the default evaluation mode, in which case the provers will try to use all the theorems in $C$ to prove the target theorem. We then call the provers by using HOL(y)Hammer's learning-based mechanism [Gauthier and Kaliszyk, 2015] of premise selection, which will first choose $n$ theorems from $C$ and then send them to the provers as premises. We also provide four additional baselines: an untrained TacticZero agent, breadth first search (BFS) and depth first search (DFS) with random choice of tactics and arguments, and always calling a single `metis_tac[]` for each theorem.

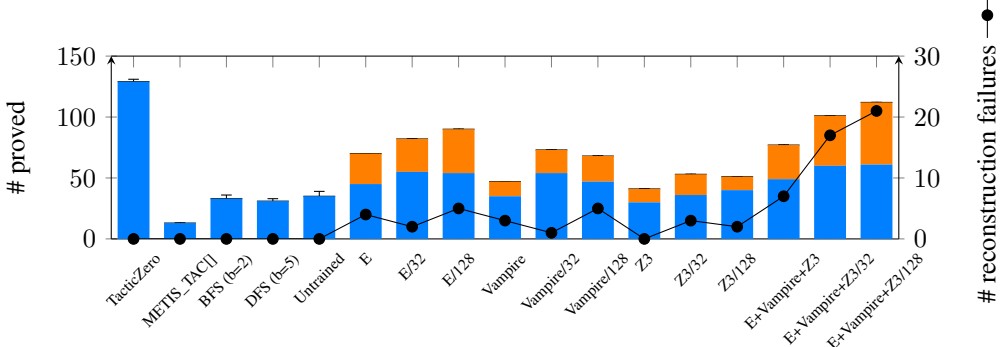

Figure 5: Performance of TacticZero compared to hammers and additional baselines. E (Z3, Vampire resp.) indicates the performance of the E prover when using all available theorems as premises. E/32 indicates the performance of E prover when using premise selection to choose 32 theorems as premises. Parameter $b$ in BFS and DFS is the branching factor that controls the number of expansions for each node. The reconstruction of proofs found by hammers is not always successful. The black line plot shows the number of failed reconstructions of each method (note different scale on right). The blue portion of each bar shows the number of theorems that are also proved by TacticZero.

Figure 5 shows the number of theorems in the testing set proved by each method given a timeout of 10 seconds. `metis_tac` as the simplest baseline can prove only 13 (out of 268) testing problems, suggesting that the majority of the testing problems are non-trivial. TacticZero solves the largest number of theorems, which is 132. In contrast, E, Vampire and Z3 solve 68, 47 and 41 theorems rescpectively, when using all the available facts as premises. The performance of hammers improves when learning-based premise selection is enabled (see E/128 for example). TacticZero is able to prove 18% more theorems (132 as opposed to 112 given by E+Vampire+Z3/128) even when we combine all the hammers together with premise selection enabled.

There is also an overlap between the theorems proved by TacticZero and those proved by other methods. For example, there are 61 theorems proved by both TacticZero and E+Vampire+Z3/128, meaning that TacticZero proved 71 theorems not found by combining E, Vampire and Z3, and the hammers working together proved 51 theorems not found by TacticZero. It is thus possible to achieve better performance by combining TacticZero and hammers in practice. We also note that learnt proofs can be dramatic improvements on what human authors wrote: Figure 6a shows an example of this.

The BFS/DFS baselines have the second worst performance among all the baselines. This suggests that proper selection of tactics and arguments is essential even when using a principled proof search strategy. In the next section, we further study the impact of different proof search strategies when proper tactic and argument policies are available.

Table 2: Ablation study for different proof search strategies in comparison with the full TacticZero. Stochastic means that when branching, the agent follows the tactic policy stochastically by sampling from the policy distribution; topk means that the agent chooses deterministically best $k$ tactics suggested by the tactic policy, where $k = b$.

| Methods | # proved | % proved | mean # timesteps | mean length of proof |
|---|---|---|---|---|
| BFS/stochastic (b=2) | 99 | 37.1 | 15.32 | 3.25 |
| BFS/stochastic (b=3) | 79 | 29.6 | 17.01 | 2.60 |
| BFS/topk (b=2) | 116 | 43.5 | 15.85 | 3.10 |
| BFS/topk (b=3) | 92 | 34.5 | 14.79 | 2.36 |
| DFS/stochastic (b=2) | 83 | 31.1 | 8.33 | 6.00 |
| DFS/stochastic (b=3) | 96 | 36.0 | 9.71 | 6.21 |
| DFS/topk (b=2) | 86 | 32.2 | 8.04 | 5.86 |
| DFS/topk (b=3) | 90 | 33.7 | 9.78 | 6.36 |
| Latest fringe | 112 | 41.9 | 11.07 | 6.94 |
| Cumulative logprob (b=2) | 113 | 42.3 | 15.61 | 3.13 |
| Cumulative logprob (b=3) | 115 | 43.1 | 15.44 | 2.38 |
| Diagonal (b=2) | 105 | 39.3 | 12.54 | 5.56 |
| Diagonal (b=3) | 103 | 38.6 | 15.27 | 4.66 |
| TacticZero | 132 | 49.4 | 11.72 | 5.00 |

## 3.3 Ablation study

We now study the proof search strategy learned by TacticZero by comparing it with other fixed search strategies including depth-first search, breadth/best-first search with different branching factors, greedily expanding the latest fringe, searching following the cumulative log probability of a goal as defined in GPT-f [Polu and Sutskever, 2020], and a diagonal search as defined in Tactician [Blaauwbroek et al., 2020]. Learned tactic and argument policies are available for all the agents in this experiment. All the agents are given the same amount of timestep budget which is 50.

The results of the experiment are shown in Table 2. The largest number of proved theorems is given by the search strategy learned by TacticZero. The BFS family tends to find shorter proofs but generally takes more timesteps to find a proof. This is because the depth of BFS is determined by the timestep budget and the branching factor[4], and there might be redundant expansion in a single level. On the other hand, the DFS family tends to find proofs quickly (as indicated by the timesteps column and row 5-8) but the proofs are rather long. This is because DFS lacks the ability to backtrack further than 1-level up in the search tree. This prevents the agent finding alternative and potentially shorter derivations by restarting the proof at a higher level when it is stuck with a particular node. Greedily expanding the latest fringe behaves similarly to DFS, but instead of backtracking, it stays with the same node forever and queries the tactic and argument policies for alternatives until the node becomes expandable. This strategy is slightly better than the DFS family in terms of the number of provable theorems, but it takes more timesteps to find proofs, and tends to find longer proofs than those discovered by DFS.

We also note that the learned proof search strategy not only finds the largest number of proofs, but also sits in the middle of the chart in terms of timesteps and length of proof — it takes fewer timesteps than a BFS agent to find a proof, and finds "deeper" proofs that may not be discoverable by a BFS agent within the same timestep budget, and tends to find proofs that are shorter than those found by a DFS agent. In fact, the learned strategy is often neither BFS nor DFS. See Figure 6b for an example.

## 4 Related Work

**Automated theorem proving (ATP)** The problem setting of machine learning for ATP is rather different from that of ITP. In ATP, one usually works with a "backbone" algorithm such as a

---

[4]For example, with a timestep budget = 50 and a branching factor = 2, the maximal depth of the search tree would be 5, which means that the maximal length of proofs that can be found by such an agent is 5.

```
Theorem EVERY2_DROP:
  ∀R l1 l2 n. EVERY2 R l1 l2 ⇒
    EVERY2 R (DROP n l1) (DROP n l2)
Proof
  Induct_on 'n'
  >- (strip_tac > fs[])
  >- (Induct_on 'l1' >- fs[]
        >- (rpt strip_tac > fs[]))
QED
(*
  (* Original human proof *)
  rpt strip_tac > IMP_RES_TAC LIST_REL_LENGTH
  > Q.PAT_ASSUM 'LIST_REL P xs ys' MP_TAC
  > ONCE_REWRITE_TAC [GSYM TAKE_DROP] > rpt strip_tac
  > ONCE_REWRITE_TAC [TAKE_DROP] > Cases_on 'n ≤ LENGTH l1'
  >- metis_tac [EVERY2_APPEND,LENGTH_DROP,LENGTH_TAKE]
  > fs [GSYM NOT_LESS] > 'LENGTH l1 ≤ n' by numLib.DECIDE_TAC
  > fs [DROP_LENGTH_TOO_LONG] > rfs [DROP_LENGTH_TOO_LONG]
*)
```

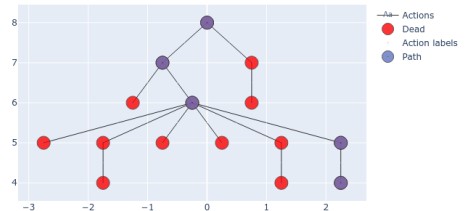

(a) A proof discovered by TacticZero. The proof has been minimized through the removal of redundant arguments provided to the `fs` tactic. The human proof of the same theorem has significantly more steps (notwithstanding its access to a much bigger pool of basic tactics).

(b) A proof search of the theorem $\forall\ x\ s.\ x \in s \Rightarrow \forall f.\ f(x) \in \mathtt{IMAGE}\ f\ s$. Red nodes represent the fringes that never lead to a successful proof, and blue nodes consist of a path from which a proof can be reconstructed. This proof search is neither a BFS nor DFS, but a unique strategy involving backtracking. See supplementary materials for an interactive HTML version of this figure.

Figure 6: Example proofs and proof search performed by TacticZero.

resolution-based or connection-based algorithm [Letz et al., 1994], upon which machine learning approaches [Urban et al., 2008, Kaliszyk and Urban, 2015a, Wang et al., 2017, Rocktäschel and Riedel, 2017, Loos et al., 2017, Selsam et al., 2018, Piotrowski and Urban, 2018, Kaliszyk et al., 2018, Crouse et al., 2019, Chvalovský et al., 2019, Zombori et al., 2019, 2020, Lederman et al., 2020, Firoiu et al., 2021] are built to guide proof search. ITP systems do not use such a "backbone" algorithm as their main framework for theorem proving. For this reason, the action space of ATP is different from that of ITP, and machine learning approaches developed for ATP are not compatible with ITP. One the other hand, proofs in ATP are often represented in a low-level language and hard to interpret as high-level mathematical concepts, in contrast to ITP which uses tactics and embodies more human-like mathematical reasoning.

There are also approaches that work by interfacing ATP with ITP systems. These tools are called *hammers* [Paulson and Blanchette, 2010, Kaliszyk and Urban, 2014, 2015b, Gauthier and Kaliszyk, 2015, Czajka and Kaliszyk, 2018]. These tools work by translating goals in an ITP system to the languages in ATP systems. If a proof is found by an ATP system, the tool then tries to reconstruct the corresponding proof in the ITP system. This process, however, might fail in the situations that the derived proof cannot be fully translated back to the ITP high-level representations.

**Interactive theorem proving** Machine learning for ITP are more closely related to our work, and most focus on supervised learning from existing human proofs in the library of an ITP system. *TacticToe* [Gauthier et al., 2020] chooses tactics based on recorded human proofs in the library, and uses a distance-weighted $k$ nearest-neighbour classifier [Dudani, 1976] for lemma selection [Kaliszyk and Urban, 2013]. *GPT-f* [Polu and Sutskever, 2020, Han et al., 2021] learns from proofs in the METAMATH [Megill and Wheeler, 2019] (LEAN [de Moura et al., 2015] resp.) library, and uses transformer-based [Vaswani et al., 2017] language models to predict proof steps and tactics. For proof search, they both use BFS as a fixed search strategy. In contrast, our approach learns a dynamic proof search strategy that enables backtracking. *GamePad* [Huang et al., 2019] and *ASTactic* [Yang and Deng, 2019] are both learning environments for the COQ theorem prover [Coq Development Team, 2004] and focus on learning from human proofs. *GamePad* targets specific sub-tasks of ITP such as the algebraic rewriting problem. *ASTactic* is more general and comes with a deep learning model that learns from human proofs to generate tactics by expanding abstract syntax trees. A beam search is then implemented as the proof search strategy to find proofs. *TacTok* [First et al., 2020] improves *ASTactic* by learning also from partial proof scripts, but follows *ASTactic* by using the same beam search as its search strategy. *ProverBot9001* [Sanchez-Stern et al., 2019] learns from proofs in the *CompCert* [Leroy, 2009] project of COQ, and uses recurrent neural networks to predict arguments for tactics. For proof search, *ProverBot9001* expands their search tree by DFS.

There are also approaches including reinforcement learning components. *HOList/DeepHOL* [Bansal et al., 2019, 2020] trains a proof guidance model to prove theorems in the HOL LIGHT theorem prover [Harrison, 1996] by continuously expanding training data. If a proof is found, it is used to generate additional training data, which is used to update the model used for exploration. Although the process is referred to as a reinforcement learning loop, it uses pre-engineered scoring for premise selection to help find new proofs, and a BFS strategy to find proofs. In our framework, the agent learns arguments (premise) selection as well as tactic selection without pre-engineered scoring, and manages proof search by itself, all through deep policy gradients.

## 5    Conclusion

We introduced a reinforcement learning framework to learning interactive theorem proving in HOL4. Rather than sequentially searching for a HOL4 proof, our agent exists within the more abstract context of our Markov decision process environment. This environment supports learning tactic prediction as well as proof search strategies in a principled and effective manner without relying on limited human examples in the libraries. We also hope that the MDP formulation opens the possibility of applying other well-developed RL algorithms to ITP. In the future, we plan to overcome the limitation of our framework by allowing free generation of complex HOL4 terms as arguments to obtain a more refined action space as well as training the agent on individual HOL4 projects such as the CakeML [Kumar et al., 2014] library.

## Acknowledgments and Disclosure of Funding

We thank the members of Prague automated reasoning group for helpful discussion on an earlier version of this work. In particular, we would like to thank Josef Urban and Thibault Gauthier for their valuable feedback on the prototype of our agent.

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
