# OpenReview forum: "TacticZero: Learning to Prove Theorems from Scratch with Deep Reinforcement Learning"
_NeurIPS.cc/2021/Conference — NeurIPS 2021 Poster_

### Official Review · Reviewer_i1kh · 2021-07-08

**Rating:** 6
**Confidence:** 4

**Summary:**

 The paper implements a reinforcement learning framework for HOL4 interactive theorem prover based on the Markov decision process. The performance is compared with existing automated theorem provers.

**Ethical Concerns:**

 I find no ethical concerns.

**Limitations And Societal Impact:**

I find no negative societal impact.

**Main Review:**

To my knowledge, the paper is the first reinforcement learning framework for general theorem proving tasks in HOL4, that focuses on argument inference. The experiments are well performed and clearly show the improvements compared to HOL4 hammer. However the comparison with the current version of TacticToe might be performed in more detail. The paper is a good attempt at reinforcement learning on mainstream proof assistants.

I wonder why the paper chooses the specific nine tactics for prediction. Is it because they are the most used tactics? The framework may not perform well on the theorems whose proof in the library contains tactics other than the nine. Are there better choices of tactics such that the framework can handle more general cases?

TacticToe does not handle the selection of HOL4 terms as arguments ==> TacticToe has a conference version and a journal version. The journal version can predict arguments while the conference version cannot. However, you cite the journal version.

The paper uses "interactive theorem-proving" several times, but I never see such usage.  "Interactive theorem proving" is more common.

**Time Spent Reviewing:**

3

---

> ### Author Response · Authors · 2021-08-10
> **Response to Reviewer i1kh**
>
> Thank you for the supportive review!
>
> Yes, the nine tactics used were chosen to reflect current usage patterns by humans; these tactics are not only extremely common in existing script files, but were chosen to be as close as possible to “complete” (being able to in principle prove all provable goals), while not being too numerous (thereby multiplying the size of the action space unnecessarily), and they have a good coverage of major argument types.
>
> A good addition would be tactics that take arbitrary HOL4 terms, for example, "suffices". However, as noted in our Conclusions section, the current framework does not handle the free generation of arbitrary terms (e.g., a novel lemma dreamed up by user), and so such tactics are not presently used by TacticZero.
>
> A more detailed comparison with TacticToe was not feasible as TacticToe currently cannot be configured to train on a given set of theorems other than an entire theory file(s) in the library. (Training TacticToe on the theories that cover the training set would then give TacticToe the advantage of knowing directly the proofs in the test set.) The authors of TacticToe have also confirmed this limitation. Nonetheless, TacticToe and the premise selection mechanism of hol(y)Hammer use similar algorithms and share the same machine learning library developed by the same authors. We hope that the comparison with hammers using premise selection (powered by hol(y)Hammer), which is included in our experiments, provided some insights.
>
> - "The journal version can predict arguments..."
>
>   Thanks for the correction! We will remove the sentence in the next version of the paper.
>
> Thank you for the review!

---

### Official Review · Reviewer_yQw9 · 2021-07-20

**Rating:** 8
**Confidence:** 5

**Summary:**

TacticZero proposes a reinforcement learning approach to theorem proving in interactive theorem provers (HOL4 in this paper, but the approach could be applied to others). The RL agent selects both a node in the search tree to expand and a proof tactic to apply in this state. This means the RL agent learns to generate good proof steps, but also learns how to search effectively. The paper formalizes this process as an MDP and demonstrates experimentally, that we can learn good search strategies in this way (compared to various BFS and DFS algorithms). The paper also shows that their approach compares favorably to existing theorem provers (not relying on deep learning). The RL agent is trained with policy gradient.

**Limitations And Societal Impact:**

See main review.

**Main Review:**

The main novelty is that the RL algorithm does not only learn what a good proof step is, but also learns a search strategy. The experimental evaluation is relatively strong, demonstrating not only an improvement over BFS and DFS baselines (which have been used until recently), but also good absolute performance.

My only critique is that the evaluation misses the comparison to a pretty obvious baseline: selecting the fringe states by the cumulative log probability, as used in GPT-f. If this baseline was present this would make me even more convinced of the approach.

The paper is very well written, I found it a pleasure to read. Some minor comments:
- lines 5ff: "This structure allows us ..." This sentence makes it sound as if your algorithm has an explicit backtracking component. However, as I understand it, your algorithm simply selects a node in the search tree to expand in each step. I think this description can be improved.
- line 9: The emphasis on the comparison to "existing theorem provers available in HOL4" is a bit besides the point. Other neural theorem provers have shown that already. In my view, the main contribution in this paper is that the search strategy is learned as well. So the emphasis of the evaluation maybe should be the comparison to other search strategies.
- line 38: I'm confused what the authors refer to. Existing neural theorem provers are not limited by the IO. The time to run inference and the time to run the proof tactics dominates all other factors.
- Figure 5: could you also list the probabilities, not only the total number of proved statements?

**Time Spent Reviewing:**

4

---

> ### Author Response · Authors · 2021-08-10
> **Response to Reviewer yQw9**
>
> Thank you for the supportive comments! We are thrilled to hear that the paper is pleasant to read.
> - "... selecting the fringe states by the cumulative log probability, as used in GPT-f"
>
>   Thanks for the suggestion! We implemented the proof search using the cumulative log probability, as described in GPT-f; the results, comparable to our table 2 and section 3.3, are as follows:
>
>   |                 | # proved | mean # timestep | mean length of proof |
>   |-----------------|----------|-----------------|----------------------|
>   | log\_prob (b=2) |      113 |           15.61 |                 3.13 |
>   | log\_prob (b=3) |      115 |           15.44 |                 2.38 |
>
>   While the GPT-f search strategy performs relatively well, it is not as good as the search strategy learned by TacticZero. We will add the above to the final paper, and fix the minor issues you commented about, thanks for the suggestion.

---

### Official Review · Reviewer_ah5U · 2021-07-31

**Rating:** 9
**Confidence:** 4

**Summary:**

This paper presents a “hammer”-style ATP for the HOL4 theorem prover. Theorem proving is characterized as a Markov decision process in which states are sequences of subgoals that arise during theorem proving and in which the action space is defined over tactics (which can take either theorem labels or terms as input). Within this setting, the paper introduces a reinforcement learning algorithm that uses a novel architecture specific to the ATP task. The proposed algorithm out-performs state of the art ATPs. A core claim of the paper is that the system is capable of backtracking during proof search.

**Limitations And Societal Impact:**

I agree with the author's assessment that "We don’t think the automation of theorem proving at its current level has potential negative societal impacts."

**Main Review:**

Overall, the paper addresses and important problem, is well-written, and contributes several novel and significant ideas.

The onerous burden placed on users of interactive theorem provers has hindered their adoption by working mathematicians and software engineers as actually useful assistants in the construction of proofs/verification of software. The success of deep reinforcement learning on other very large search problems, such as two player games, suggests that RL could contribute significantly to the usability and utility of interactive theorem provers by automating a significant amount of the proof construction and proof search processes. However, as the Related Work section of this paper discusses, existing approaches either focus on specific supervised tasks or else lack capabilities that are intrinsic to any non-trivial search process (e.g., backtracking).

The paper takes several well-established ideas from DRL and adapts them to the ATP domain:

1. Structuring the state space of the MDP in a way that is amenable to backtracking during proof search.
2. Replay in order to deal with sparse rewards (“fringes”).
3. Reward shaping to discourage local optima.

Although these ideas are already well-studied in the DRL literature, effectively adapting each to the ATP domain in a way that actually out-performs existing ATP methods is a highly non-trivial contribution.

In addition to adapting existing ideas from DRL, the paper also contributes a novel architecture with several features that are relatively unique to ATP:

1. a transform-based autoencoder for HOL terms,
2. fringe (e.g., subgoal), tactic, and argument selection networks, and
3. a clever definition of policy distribution in terms of both subgoal and tactic selection.

The empirical evaluation of the paper is sound — the system is compared against state of the art hammer-style provers, including some that contain machine learning components, and an ablation study validates the primary claim of the the paper (that TacticZero is effectively learning to backtrack during proof search). I scrolled through the theorems used for the empirical evaluation and believe these are a reasonable set of benchmark problems for evaluating hammer-like systems. None-the-less, I wonder if papers on this topic should start using a standardized set of problems (a la CASC).

The most substantial weakness of this submission is the lack of source code combined with insufficient material in the appendix to easily reproduce the system and relevant experiments.

I recommend accepting this paper because it addresses an important problem, contributes and clearly describes several important conceptual insights, and presents promising empirical evidence that these insights, combined with relatively straightforward adaptations of established ideas from DRL, are capable of competing with the state of the art in ATP for higher-order theorem proving.


**Time Spent Reviewing:**

3

---

> ### Author Response · Authors · 2021-08-10
> **Response to Reviewer ah5U**
>
> Thank you very much for the supportive comments and recognizing our contribution!
>
> We will include more details in the appendix of the final paper. We are of course also highly motivated to open source the project once we obtain permission from our organisation, which should pose no problem.

---

### Official Review · Reviewer_Sr8m · 2021-08-02

**Rating:** 6
**Confidence:** 4

**Summary:**

This paper discusses how to train a tactic-based theorem prover from
scratch with a limited set of tactics and no supervised learning data.
One particularly interesting feature is that TacticZero can teach itself
strategies to guide the search.


**Limitations And Societal Impact:**

Yes.

**Main Review:**

The connection between the MDP and tree search is not explicit enough in the paper. Some motivation could be provided along the following lines.
Tactic-based theorem proving is naturally understood as a tree search problem where actions are tactics and states are sets of goals.
In the reformulation, states are the search tree (set of fringes) and actions update the search tree (choose a node and apply a tactic).
This way, the agent can backtrack in the search tree using during a single episode (linear sequence of actions).

A drawback to this linear approach (compared to the MCTS approach in RLcop) is that this does not explicitly distinguish between actions that were necessary for the final proofs and actions that were not.
For example, an action that produces a new fringe that is irrelevant for the final proof will receive a reward of 0.1 at step 1 and 5.0 at step n if the top goal is proven. I would expect that when a proof is found, such irrelevant actions would be removed (or re-evaluated) before training but this does not seem to be the case.

In the experiments, outperforming ATPs/hammers is a strong point that validates the approach presented here and the ablation study shows that the search strategy developed is better than DFS and BFS. The comparison could have included a slightly more advanced strategy that alternates between BFS and DFS such as the one used in Tactician (should be cited). Also, higher-order ATPs have been doing very well recently - see the GRUNGE evaluation (done on HOL4) and the two latest CASC THF competitions.  And TacticToe seems a natural candidate to compare with.

Overall, the approach presented here is interesting as it is conceptually simple and shows how one can learn to backtrack using the REINFORCE algorithm which is of general interest.

The experiments demonstrate learning proof strategies for tactic-based theorem proving from scratch on a thousand carefully selected theorems but the usability of TacticZero on larger developments remains to be demonstrated.


Minor:
> Large action space

How large?

> After selecting a fringe, by default we select the first goal in that fringe to work on, because all of the
     goals within a fringe have to be proved in order to prove the main goal, and the order in which they
     are proven is irrelevant.

Although all goals need to be proven at some point, the choice of a goal would influence the search by producing different fringes and therefore would affect fringe selection.

> holyHammer

HOL(y)Hammer

> Background on RL-based approaches:

A number of RL (Dagger-style) based AI/TP systems are missing, starting with MaLARea and ATPBoost, plcop/graphcop, ENIGMA, etc.

=========

Update after the author response:

Thanks to the authors for their replies and additional experiments. I have decided to increase my score, even though I would much prefer to see the evaluation on all of HOL4.

I am quite unconvinced by the answer on comparisons to other HOL4 systems, but my impression is that running their system on all of HOL4 may be too expensive for the authors.

=========







**Time Spent Reviewing:**

4

---

> ### Author Response · Authors · 2021-08-10
> **Response to Reviewer Sr8m**
>
> Thank you for the review!
> - "Some motivation could be provided along the following lines..."
>
>   Indeed, states intuitively represent the search tree (set of fringes). We did not make this explicit, because these sets themselves do not have a tree structure until we add back the information about actions connecting the fringes, and so we tried to keep the description formal to avoid possible confusion. We will rephrase the section following your suggestion, and make the motivation more clear.
>
> - "A drawback to this linear approach ..."
>
>   It is true that policy gradient by its nature does not explicitly distinguish between actions that were necessary for the final proofs and actions that were not. However, if the agent learns to collect more rewards by making useless progress, then the overall proof attempt becomes more likely to fail due to the finite episode length. The corresponding negative terminal reward in turn has the effect of discouraging such behaviors.
>
>   On the other hand, this can also be mitigated by shaping the rewards differently. For example, it is possible select the fringes that do not directly contribute to the final proof and adjust the rewards as you have suggested. Intuitively, our proposed reward shaping mitigates issues of otherwise overly sparse rewards, and by rewarding "useless" progress encourages exploration.
>
> - "The comparison ... slightly more advanced strategy ... the one used in Tactician (should be cited)"
>
>   Thanks for the suggestion! We implemented the algorithm (called "diagonal" search in the Tactician paper) which is a skewed form of breadth-first search that explores the subtree starting from promising tactics deeper than those starting from less promising ones; the results, comparable to our table 2 and section 3.3, are as follows:
>
>   |                | # proved | mean # timesteps | mean length of proof |
>   |----------------|----------|------------------|----------------------|
>   | diagonal (b=3) |      103 |            15.27 |                 4.66 |
>   | diagonal (b=2) |      105 |            12.54 |                 5.56 |
>
>   While the diagonal search performs relatively well, it is inferior to the search strategy learned by TacticZero. We will add the above to the final paper, thanks for the suggestion.
>
> - ".. GRUNGE evaluation ... CASC THF competitions"
>
>   We agree that papers on this topic should start using a standardized set of problems, which is also mentioned by reviewer ah5U. However, this is not yet feasible due to the incompatibility between various ITP systems. Nonetheless, the dataset we used approximates GRUNGE as they both come from HOL4's core library without manual cherry-picking (see supplementary materials), and we believe that the current experiments have demonstrated the effectiveness of our novel approach, which is the main purpose of this paper.
>
> - How large?
>
>   Let the length of the list of arguments be n and the average size of the set of candidate theorems be k. The action space is then at least k^n, where k=486 and n=5 in our settings. A more detailed description can be found in section 2 of the supplementary material.
>
> - "... the choice of a goal would influence the search ..."
>
>   Yes, what is used in the paper is a simplified design. One could just use the learned scores to choose goals. In early experiments, we did not notice significant difference in performance between the two designs.
>
> We will add the missing references into the paper. Thank you very much for the comments!

---

### Decision · Program_Chairs · 2021-09-27

**Decision:**

Accept (Poster)

**Comment:**

This paper presents an RL approach to learning how to construct proofs in the context of an interactive theorem prover. Unlike prior work, the approach presented in this paper does not rely on existing human generated proofs. The main contribution of the paper is in the way it sets up the learning problem, although the learning techniques are fairly conventional.

The main shortcoming of the paper is that it does not compare with other learned theorem provers. However, the non-learning-based baselines that it does compare against are state-of-the-art. I was actually surprised that the BFS search strategies were able to prove so many more theorems than the off-the-shelf hammers (Z3, E, Vampire).

I suspect there is still a lot of room for improvement in this space, but this paper makes a clear and solid contribution to the state of the art in learned theorem provers and should be accepted.